# Innovative and Sustainable Food Production and Food Consumption Entrepreneurship: A Conceptual Recipe for Delivering Development Success in South Africa

**Faith Samkange** [1,*], **Haywantee Ramkissoon** [1,2,*], **Juliet Chipumuro** [3], **Henry Wanyama** [4] **and Gaurav Chawla** [5]

1   Centre for Contemporary Hospitality & Tourism & Centre for Business Improvement, College of Business, Law, & Social Sciences, University of Derby, Derby DE22 1GB, UK

2   College of Business & Economics, Johannesburg Business School, University of Johannesburg, Johannesburg 2006, South Africa

3   School of Hotel Management, Stenden University, Saint Alfred 1142, South Africa; juliet.chipumuro@stenden.com

4   Tshama Green Consultants, Johannesburg 2006, South Africa; Tashmaconsult@gmail.com

5   South Wales Business School, University of South Wales, Newport NP20 2BP, UK; gaurav.chawla@southwales.ac.uk

*   Correspondence: F.Samkange@derby.ac.uk (F.S.); H.Ramkissoon@derby.ac.uk (H.R.)

**Abstract:** Innovative food production and food consumption entrepreneurship can be viewed as a recipe for delivering sustainable development goals to promote economic, human, and community growth among vulnerable and marginalised communities in South Africa (SA). This study critically analyses the trends and related issues perpetuating the development gap between privileged and marginalised communities in SA. It explores the link between innovative food production and food consumption entrepreneurship and underdevelopment based on sustainable development goals (SDGs). The study also generates a conceptual model designed to bridge the development gap between privileged and marginalised communities in SA. Philosophically, an interpretivism research paradigm based on the socialised interpretation of extant literature is pursued. Consistent with this stance, an inductive approach and qualitative methodological choices are applied using a combination of thematic analysis and grounded theory to generate research data. Grounded theory techniques determine the extent to which the literature review readings are simultaneously pursued, analysed, and conceptualised to generate the conceptual model. Research findings highlight the perpetual inequality in land distribution, economic and employability status, social mobility, gender equity, education, emancipation, empowerment, and quality of life between privileged and marginalised societies in SA. Underdevelopment issues such as poverty, unemployment, hunger, criminal activities, therefore, characterise marginalised communities and are linked to SDGs. Arguably, food production and food consumption entrepreneurship are ideally positioned to address underdevelopment by creating job opportunities, generating income, transforming the economic status, social mobility, and quality of life. Although such entrepreneurship development initiatives in SA are acknowledged, their impact remains insignificant because the interventions are traditionally prescriptive, fragmented, linear, and foreign-driven. A robust, contextualised, integrated, and transformative approach is developed based on the conceptual model designed to create a sustainable, innovative, and digital entrepreneurship development plan that will be executed to yield employment, generate income and address poverty, hunger, gender inequity. To bridge the gap between privileged and marginalised societies. The conceptual model will be used to bridge the perpetual development gap between privileged and marginalised societies. In SA is generated. Recommended future research directions include implementing, testing, and validating the model from a practical perspective through a specific project within selected marginalised communities.

**Keywords:** sustainability; food production and food consumption entrepreneurship; privileged and marginalised communities; community; human and economic development

## 1. Introduction

Entrepreneurship failure in South Africa (SA), where unemployment continues to increase, is perpetuating inequality and accelerating the development gap between the marginalised and privileged societies [1–3] The purpose of this study is to create a contextualised understanding of the human, economic, and community development trends perpetuating the development gap between the privileged and marginalised societies in SA. The study explores ways of generating a sustainable, innovative entrepreneurship development model that can effectively address the underdevelopment and inequality issues such as unemployment, poverty, hunger crime among vulnerable people. Based on the current unemployment rate of 32.6 [4] the link between human, social and economic development with entrepreneurship remains critical in SA. Entrepreneurship, especially in marginalised communities, is designed to create jobs, improve the economy and the quality of life. Unfortunately, more than 70% of entrepreneurship initiatives in SA continue to struggle to imply there is some insignificant impact of such initiatives on development [2–4]. Effectively, managing entrepreneurship development failure is likely to bridge the economic, human, and community development gap between these societies. This is crucial in a country where 25 years after the abolishment of apartheid, inequality between privileged, vulnerable, and marginalised communities is perpetual and worsening in the COVID-19 pandemic era [5–7]. Research has exposed prescriptive, fragmented, linear, foreign-driven and traditional enterprise business development approaches as major factors impeding success [8,9]. Based on these arguments, innovative food production and food consumption enterprise interventions should challenge the unproductive traditional strategic and operational processes and procedures, which continue to focus primarily on the fragmented and isolated articulation of issues [6,8,10,11]. Community-driven and successful enterprise initiatives should, in reality, address the underdevelopment issues perpetuating apartheid values and principles within the marginalised communities. This underdevelopment is demonstrated by high levels of inequality in land distribution, economic status, employment and employability status, social mobility, gender equity, education, empowerment, and quality of life [5,12,13]. Managing these issues demand an integrated practical and grassroots-level approach. Entrepreneurship research initiatives rarely target underdevelopment in vulnerable communities in SA with a stipulated focus on sustainability and identified sustainable development goals (SDGs) as suggested by the United Nations [14]. A customised practical-oriented development approach within marginalised societies is yet to be fully explored; hence, the levels of inequality remain perpetual. Consequently, an eradication of the development gap between marginalised and privileged societies in SA is yet to be registered [15,16]. This study creates an opportunity to articulate this perpetual gap and seeks to develop a contextualised solution from a conceptual perceptive using grounded theory [17]. A contextualised and integrated perception of the link between marginalisation and underdevelopment trends characterising sustainability, innovative entrepreneurship, human, social and economic development in SA is pursued. A theoretical model reflecting this perception while generating contextualised and grassroots solutions to the high levels of underdevelopment and inequality is generated.

## 2. Research Background

South Africa as a nation has a population of 30.3 million, and 55.5% of the population lives in poverty [4]. Food production is a major part of agricultural entrepreneurship in SA, which is designed to play a huge role in addressing underdevelopment issues such as poverty, especially among the marginalised communities. Linking entrepreneurship, economic development, job creation, and human development in SA is reflected in the national development plan. According to the plan, small to medium enterprises (SMEs) should constitute 60–80% of the gross domestic product (GDP). Strategically, the plan aims to address unemployment, which continues to rise. This is not surprising because (SA) ranks 114 out of 189 countries for human development. Futshane [18] and Stas SA [4] believe that the human development index should be perceived as a contextualised

statistical stipulation of levels of education, income, quality of life, life expectancy, etc. Based on this perception, a recent survey on the impact of the COVID-19 pandemic concludes that SA is one of the most unequal countries in the world [18]. Inequality in SA also highlights gender development issues that are based on sexuality profiles. The extent to which women are respected and have access to equal opportunities for human development compared to men leaves a lot to be desired in SA. This is justified by a gender development index of 0.986 in SA [1]. Supporting this argument, Stats SA [4] and Oxfam [18] show that 55.5% of the SA population, mostly women, live in poverty. An average woman in SA earns 30% less than men. Further analysis associated with human development demonstrates that the current employment situation indicates an increase in job losses as a result of the COVID-19 pandemic. This unprecedented COVID-19 pandemic has recently registered 2.52 million cases and 74.623 deaths. It is agreed that more cases are registered in marginalised communities in SA, further exposing development inequalities [4]. Consistent with this argument, recent research indicates that vulnerable indigenous people are living in overpopulated urban townships and underdeveloped rural areas where poor health remains an issue due to poverty, poor water supply, and poorly structured health care facilities [18]. Further advancing this argument, Al-Omoush et al. [11] articulate the need to collaborate innovative ideas as social capital with business development to address in response to the COVID-19 crisis in their recent research study. In an effort to address all these emerging development concerns and implement some of the transformative research suggestions, SA has embarked on community and economic development strategies such as entrepreneurship and expanded public works programmes [5]. This is designed as part of the national economic strategy. A marked increase in entrepreneurship activity, especially in privileged societies, has since been registered [6–9]. However, more than 70% of these business enterprises continue to fail. The COVID pandemic is exacerbating the situation making the impact of such programmes insignificant, especially within vulnerable communities [2–4].

Using extant literature, this paper explores these underdevelopment and entrepreneurship issues to create an integrated solution towards bridging the human, social, and economic development gap between privileged and marginalised societies in SA. Extant literature review, according to Sangwan et al. [19] and Yarwood-Ross and Jack [20], is linked to the grounded theory research approach. Implied by these scholars is the fact that a range of published research articles related to the objectives under investigation is used as a basis for data collection. It can also be argued that extant literature review is linked to content analysis, which is applied in this study. The study objectives, therefore, seek to:

- Analyse the human, economic and social underdevelopment entrepreneurship trends within the vulnerable and marginalised communities in SA;
- Explore the link between innovative food production and food consumption entrepreneurship and the identified underdevelopment issues within the vulnerable and marginalised communities in SA based on sustainable development goals;
- Generate a contextualised and integrated conceptual model to facilitate a grassroots approach towards sustainable food production and food consumption entrepreneurship designed to bridge the human, social, and economic development gap between privileged and marginalised communities in SA.

The conceptual framework applied in this study to generate the outcomes identified is reflected in Figure 1.

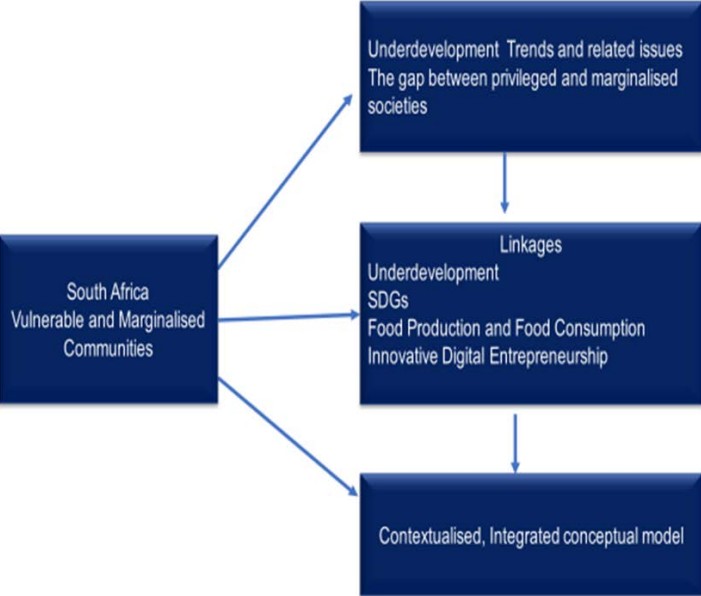

**Figure 1.** Conceptual framework.

The study contributes to knowledge through a conceptual model that is designed to promote an integrated perception of the community, human and economic development issues at stake in SA. It also demonstrates how innovative and sustainable food consumption and food production entrepreneurship can facilitate bridging the development gap between privileged and marginalised communities in SA. The study argues that the conceptual model contributes towards grassroots development initiatives. This is based on the argument that an integrated multidisciplinary perception of the complex issues at stake [5,15,16] and a wide range of expertise can be deployed as a recipe for enterprise development success within marginalised communities. The model is designed to create the much-needed impact desired to address the perpetual underdevelopment that currently characterises these communities despite the end of apartheid in SA [2,3,5].

## 3. Research Methodology

This study is based on grounded theoretical underpinnings to addresses the identified development, sustainable and entrepreneurship variables reflected in the research objectives. Significantly, the research is designed to generate a theoretical model that can be applied to resolve the increasing developmental issues at stake within vulnerable communities. Grounded theory principles, techniques [17,18] and related methodological processes are used.

### 3.1. Grounded Theory

Grounded theory was developed by Glaser and Strauss [17]. It is a research approach that generates new insights based on the data emerging from the study. The data creates an understanding of the issues at stake based on the codes and themes emerging. Analysis, synthesis, and evaluation of the themes then generate concepts that are ultimately used as a basis for developing theories [17,21,22]. Further advancement of the arguments on the grounded theory perception highlights the link between data collection and analysis as a combined process. Data collection is therefore pursued until a saturation point is reached. The implication is that data collection continues until no more new input is recorded [22]. When using extant literature, readings are pursued until a saturation point is reached. Saturation refers to a situation where no new data is reflected. Research articles, in this case, were consulted based on the specific areas identified in the study based on the research objectives established. Current developments on grounded theory reflect a wide range of

scenarios under which the theory is applicable [21]. Practically, grounded theory can also be applied in combination with content and thematic analysis.

### 3.2. Thematic Analysis Based on Extant Literature

Thematic analysis collects data and analyses it based on themes that then facilitate conceptualisation of findings [23] towards the generation of theories. Five themes emerged from the extant literature based on the grounded theory approach. The empirical research studies pursued to generate each theme based on the grounded theory saturation technique are reflected in Table 1.

**Table 1.** The SA research articles selected for literature review.

| Theme | Key Research Articles | Number of Articles |
|---|---|---|
| The concept sustainability | Global perceptions<br>14, 30, 31, 32, 34, 35, 36<br>Sustainability and the South African situation<br>1, 2, 3, 4, 5, 20 38, 8, 39, 27, 29, 42, 66, 67 | 21 |
| Sustainable community, human and economic development in South Africa | 4, 39, 15, 5, 8, 9, 7, 62, 49, 43, 60, 51, 53, 54, 56, 57, 88, 64, 101, 40, 41, 42 | 22 |
| Food production and food consumption | 2, 4, 5, 34, 8, 60, 70, 71, 72, 75, 66, 67, 68, 76, 77, 113, 114 | 17 |
| Sustainable food production and food consumption entrepreneurship | 37, 10, 82, 8, 38, 84, 61, 52, 53, 55, 57, 58, 60, 69, 79, 70, 81, 107, 73 | 19 |
| Innovative sustainable and digital entrepreneurship | 70, 79, 85, 86, 87, 90, 89, 92, 94, 95, 97, 105, 66, 108, 113, 96, 100, 103, 101, 102, 113 | 21 |
| Total number of key articles on South Africa | | 100 |

### 3.3. Research Process

Based on grounded theory and thematic analysis, the research process pursues the research articles reflected in Table 1 to review the extant literature and related empirical research studies. Philosophically, this conceptual paper is based on the review of readings from an interpretivism perspective. Interpretivism is a paradigm focused on subjective interpretation of issues based on human interest [24,25]. The research process reflected in Figure 1, therefore, adopted an open-minded and inductive research approach.

The inductive research approach, according to Punch [24,25], uses the emerging findings from the extant literature to create new insights. This perception implication suggests contextualised ways of presenting the insights immerging from the selected research articles. In this case, immerging insights from the study findings are presented in a personalised and subjective discussion. The ideas established from the empirical studies were then applied to create themes. Further readings were pursued to enhance the comprehension and application of the findings until a saturation point was reached, which is consistent with the grounded theory [17,21,22]. A reflective analysis of trends, linkages, propositions, and concepts emerging from the themes was thereafter used as a basis for the design of the conceptual model. This is homogenous with conceptual studies, as reflected in similar research by Sanchez-Satamria et al. [26]. The review of the research articles used a qualitative methodological stance that is appropriate for pursuing a detailed, in-depth analysis of issues. The qualitative analysis of selected research article readings is referred to as content analysis [24,25]. The grounded saturation technique justifies the established total of 100 research articles that were used to generate the qualitative data presented in the form of a discussion. A profound set of criteria for the selection

of the identified key research articles underpins the significance of the study objectives outlined, which are summarised in Figure 1. Further justification of research article selection includes effective peer review of journals written in English and based on their relevance to SA. Although other readings were picked up, these were only applied to facilitate comprehension, application, analysis, and evaluation of the arguments presented, and hence they are not reflected in the key research article sample shown in Figure 1. This is consistent with similar research conducted by Tsalis et al. [27], who argues that such a process seeks to create a customised and contextualised but open-minded perception of issues typical of qualitative research justifying the key criteria applied for the selection of articles focused on research reports related to SA [25,26]. Substantiating the qualitative analysis of the literature review, it is important to identify the themes created, which included human, social and economic development in SA, sustainable food production and food consumption entrepreneurship, innovative digital entrepreneurship development (see Table 1). Qualitative research in this respect can be argued as a study of phenomena in a contextualised situation focusing on a variety of perceptions [24]. Figure 2 reflects the application of grounded theory techniques to synthesise, evaluate and reflect upon the phenomena of marginalised societies in SA with an ultimate generation of concepts used to generate new knowledge [23,24]. In this case, the new knowledge created espouses the design of a conceptual model. Qualitative research experts refer to this as part of the conceptualisation process linked to the inductive approach culminating in the generation of grounded theory [22,24,25]. In this conceptual paper, this grounded theoretical outcome contributes to knowledge by creating a solution towards bridging the development gap between the privileged and marginalised communities in SA.

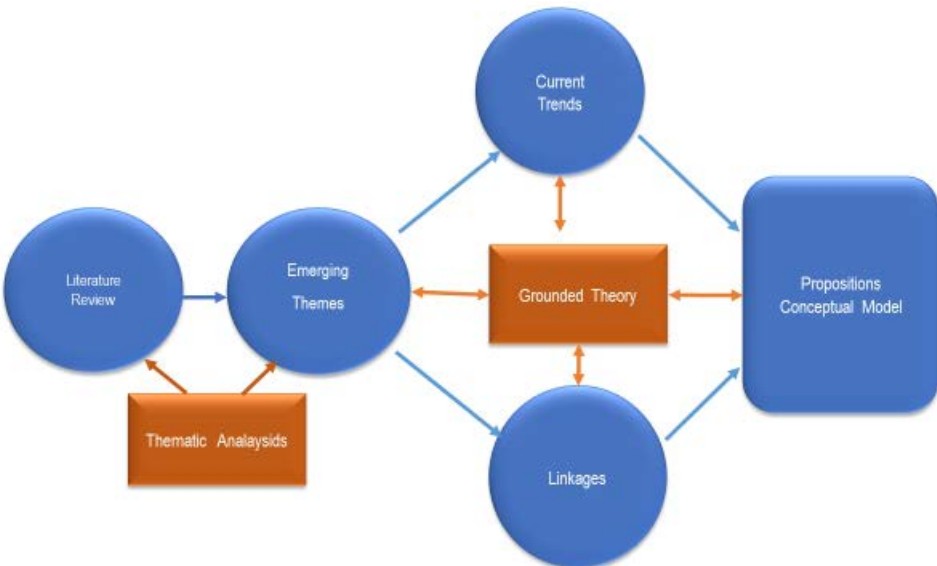

**Figure 2.** The research process. Sustainable development goals and marginalised communities.

## 4. Literature Review

### 4.1. The Concept of Sustainability

Initial perspectives on sustainability were based on the Convention on Environment and Development held in 1987. These perspectives were further articulated in agenda 21 of the Rio Conference of 1992 [28,29] and later expanded in the 2018 global convention [14]. A critical analysis and synthesis of these perceptions emphasise the eccentric development associated with the management of resources in ways that enable humans to thrive without compromising the needs of future generations and other forms of life [28–30].

Initially, three pillars or domains of sustainability were created reflecting the significance of economic, social, and environmental development with an emphasis on resource management and its impact [31]. Further developments saw a number of theoretical

models reflecting an expanded conceptualisation [32]. However, many perceptions did not include a clear set of goals to facilitate globalised implementations and accountability. Consequently, a global initiative decided to break down the diversified range of theoretical concepts and models into the SDGs that ensured a broader articulation, achievement, and assessment of sustainability [33]. These are reflected as the specified SDGs [14] (see Figure 1). Many scholars now perceive sustainability from both a strategic and operational viewpoint [34,35]. However, research on sustainable entrepreneurship business development in SA is yet to be fully explored.

Reflecting on these goals, it can be argued that no poverty (SDG1) as a goal, for instance, is synergised with quality education (SDG4), gender development (SDG5), zero hunger (SDG2), suitable health and well-being (SDG3), environment management including corporate social responsibility (CSR) and all the other goals [36]. While much debate regarding greening and management of the environment through effective use of resources and prevailing ecosystems in an effort to reduce carbon emission and preserve natural resources is taking precedence in SA (SDGs 11 and 12) [37], the significant sustainability issues in SA hinge around inequality. Research studies focusing on SA articulate inequality and underdevelopment trends in SA, highlighting the significance of economic underdevelopment and poverty, food insecurity, health and malnutrition, gender bias and lack of empowerment, limited social mobility, poor quality of life among the vulnerable communities within the marginalised societies [2,3,38–42]. This is quite consistent with the development trends in SA, where the highest levels of inequality have been recorded consistently with the SDGs (SDGs 5, 10, 16) [14,36]. A key point resonating with these research narratives indicates that the attainment of these goals highly depends on how the synergies are leveraged, emphasising the need for integrating sustainable activities and their related evaluation [33–35]. An analysis of these research articles highlights inequality associated with economic, environmental, and community underdevelopment trends raising tremendous concerns regarding the contextualised conception of sustainability [2,4,8,28,30]. The major concern emerging is that data-driven empirical research hardly demonstrates the correlation and interaction between the inequalities established and the goals. This is an area yet to be fully explored, suggesting current research trends in SA related to inequality [3,8,20,30,38,39] need to become more focused on the sustainable development goals [14,28]. Creating a contextualised perception of sustainability in SA is critical because business development, particularly food production and food consumption trends [4,43,44], are showing that sustainability is becoming a critical customer and business development profile in SA [37].

A summary of the findings under this theme highlights underdevelopment and inequality as major human, economic, and community trends. The findings also demonstrate how a wide range of SDGs (1, 2, 3, 4, 5,10,11, 12, 16, 14) are linked to these underdevelopments, inequality issues, and their consequences, and yet research addressing these issues hardly links them to the SDGs. The key proposition emerging from this narrative is:

**Proposition 1.** *Inequality trends regarding human, economic, and social development issues related to the marginalised communities in SA are directly linked to the concept of sustainability and reflected through the specific SDGs including 1, 2, 3, 4, 5, 10, 11, 12, 16, 17, (Figure 3).*

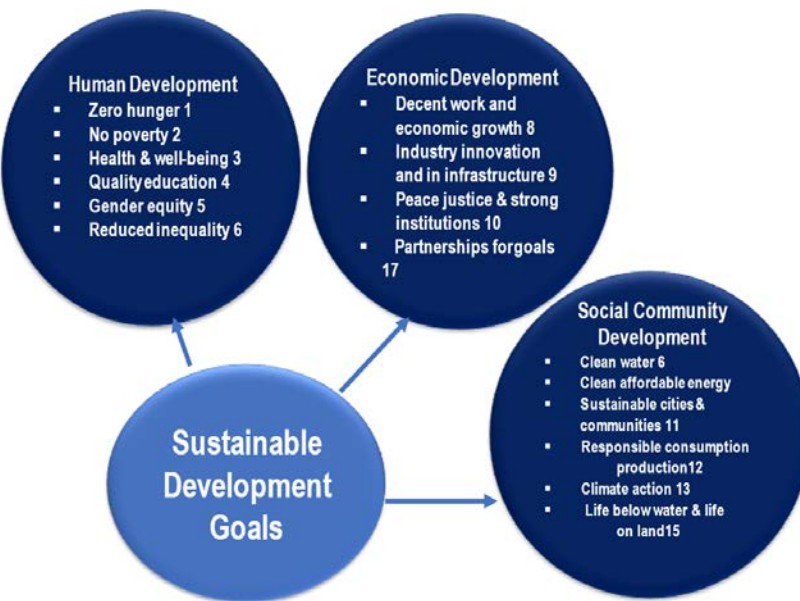

**Figure 3.** The sustainable research goals associated with economic, human, and community development trends in SA.

### 4.2. Sustainable Community, Human and Economic Development in South Africa

Based on the sustainability perceptions and the related SDGs, community, economic and human growth constitute key elements of development [14]. Dishearteningly, more than 25 years after the abolishment of the apartheid system in SA, a negative footprint of this inequality legacy remains a visible complexity expanding the development gap between privileged and marginalised communities [5]. Empirical research on human and community development in SA does not always give due attention to this gap that continues to shape high levels of underdevelopment among the disadvantaged groups of people [12,36,44]. Underplaying this issue is perpetuating inequality among the vulnerable and privileged members of the communities in SA (SDG 10). An analysis of this research reflects some critical issues. Firstly, the need to interrogate inequality and formulate poverty reduction strategies and practices emerges (SDG1). Secondly, the underdevelopment complexity associated with inequality remains prevalent in many parts of marginalised communities in SA to date [8,12] (SDG 4), and yet these communities still rely on privileged communities for jobs, advice, support, and many other things as the privileged communities continue to own the means of production [9]. Thirdly, the levels of dependence and lack of empowerment in terms of development within the marginalised societies remain significant. Based on these observations, it can therefore be argued that SA is still struggling to bring the much-desired economic development and social equity (SDG 8) needed to improve the quality of life among these communities. Supporting this argument, some scholars argue that high levels of underdevelopment will continue to highlight the lack of equality, creating a wide range of issues, including criminal activities, in different ways unless drastic action is taken immediately [6]. Current research studies in SA indicate a major resuscitation of this debate due to the current COVID-19 pandemic impacting the current fragile political environment in SA [6,7]. Many critics of inequality supporting diversity and inclusivity principles agree that perpetuating poverty, unemployment, and lack of income characterises the indigenous African vulnerable people who hardly own the means of economic production [7,45] (SDG 1, 8).

Although economic empowerment initiatives among indigenous communities such as cooperatives and related entrepreneurship are evident in SA, they have yet to make a significant impact on the quality of life within the marginalised communities (SDG 3). Development theorists suggest that initiatives towards job creation, poverty alleviation, and access to business finance need to be prioritised in order to redress the unequal distri-

bution of ownership, management, and control of South African financial and economic resources [15,46] (SDG1, 2, 3, 8, and 10). Consensus prevails regarding a downward spiral national economic growth rate that is predicted to reach −5% of the gross domestic product (GDP) by the end of 2021 (SDG 8) [15,16]. Exacerbating this negative development is the new COVID-19 DELTA variant, which has seen cases spiral to unprecedented levels, putting SA in an even more vulnerable global position [2,4] (SDG 3). Current employment and income generation initiatives have seen a very insignificant impact evidenced by a downward spiral development trend. Research indicates that over 3 million people have lost their jobs [47]. It is not surprising that this has greatly impacted the lowest educated and vulnerable members [37], with 66.65% of those affected being women, emphasising the gender inequality prevalence in SA. A research study analysing the impact of COVID-19 on South Africa by Parry and Gordon [40] articulates the inequitable gender practices and their negative impact on accelerated unemployment among women, their poor income levels and, quality of life in general. In collaboration, further arguments pursued in similar research demonstrate consensus in this narrative [48–50]. Statistical data sets support the arguments emerging from these studies by indicating that individual income levels have been reduced by 40% and household income by 70% [16] (SDG 1 and 8). A related major concern towards food security, which is evidenced by an increase of 22% of hunger cases within the vulnerable societies, is reflected in national political debates [47,51] (SDG 2). Food insecurity is rampant within marginalised societies signifying scarcity in the availability and accessibility, including affordability of food. Current gender emphatic studies conclude that women are more vulnerable to food scarcity in SA because of the gender inequalities existing [52–54]. Gender equity, including community emancipation and empowerment and accessibility to basic resources such as food, remains evasive and quite elusive for vulnerable communities [52,54], propelling the need for further research in this domain (SDG 5).

These arguments are now being used to justify the significance of sustainable human and community development through viable food production and food consumption entrepreneurship initiatives in SA [51,55,56] (SDG 11 and 12). Arguably, research efforts have articulated the significance of such initiatives in a country where poverty and inequality are perceived by many as a national disaster [14,46,57], and yet not much has yet to be achieved.

An analysis of the underdevelopment issues emerging in SA based on the sustainability domains includes a broad range of environmental, economic, and social development issues exposing poverty alleviation, gender development, inequality, climate change, employment creation, economic development, food security, community development, and human well-being as key problems [8,56] (SDG 1, 3, 5, 8, 11, 9, 16) A critical examination of a number of research articles demonstrates a new prevalence of globalised strategic and operational policy development plan and activism on sustainability [31,58–62]. It is also clear that consensus among the research proponents of development now prevails regarding the fact that marginalised communities' real development needs and interests in SA are yet to be understood [45,59,60,63]. Consequently, the impact of development interventions among these communities remains highly inconspicuous, according to recent research studies [41,43,64,65], extending the gap between local marginalised indigenous communities and the privileged minority communities in the post-apartheid period and more so during the current COVID-19 pandemic era in SA [4,8,40,66] (SDG10).

Many studies conclude that the apartheid legacies of underdevelopment and inequality within the marginalised communities remain the major human, economic, and community underdevelopment trend, which is a big concern [8,45,67].

Based on this trend, it is obvious that the greatest challenge facing SA is how to create dynamic and innovative strategies to leapfrog cultural and socio-economic transformation towards equality, emancipation, and empowerment of vulnerable communities [62,68] (SDG 11). An analysis of research on the development trends established in SA reflects the need to reduce inequality between the privileged and marginalised communities by

addressing the complex development issues that are linked to the SDGs [14]. The arguments presented under this theme highlight the underdevelopment issues impacting SA and the need to bridge the development gap between the vulnerable people and the privileged minorities in SA. This generates the following propositions.

**Proposition 2.** *The human, community, and economic underdevelopment trends, especially inequality within marginalised societies in SA, are linked to the following SDGs: 1, 2, 3, 4, 5, 7, 8, 9, 10, 11, 12, 13, 16 (Figure 1).*

**Proposition 3.** *There are many SDGs linked to inequality and the related underdevelopment issues demonstrating the complexity associated with the development gap between privileged and marginalised societies in SA.*

### 4.3. Food Production and Food Consumption

Efforts to address some of the trends and issues associated with the vulnerable and marginalised communities in agricultural food production and food consumption entrepreneurship should play a pivotal role towards development in SA [3,42,44,69,70]. In a study focusing on nutritious food production and consumption in rural areas of SA (Hendriks et al.) [69] articulates the link between food production, food security issues, and sustainability. The study findings and conclusions highlight the risk of hunger and food insecurity within marginalised societies in SA. Conceptually, technical agricultural food production and food consumption, community, human development, and economic processes resonate with food security issues [41,44,69,71]. Food security is a major sustainable development challenge that defines and distinguishes the developing economy of SA from developed countries, according to a study conducted by Queenan et al. [42]. SA remains one of the poorest countries in the world [41,44,69]. Ideally, the levels of availability, accessibility, and affordability of food within vulnerable communities is a big food security issue in SA. Advancing the arguments raised by Queenan et al. [42] in research addressing food consumption patterns and sustainability, other scholars argue that socially, physically, and economically food security is designed to combat hunger and malnutrition, which are major development problems affecting health and well-being in SA [2,72–75]. Supporting these research arguments, FAO confirms that the high levels of food insecurity are a major challenge to SA, especially within vulnerable communities [11,76,77]. The lower levels of human, community, and economic development in SA explain why food insecurity remains topical and problematic [42,44,69]. Based on this argument, it is agreed that high food production activities are consistent with high levels of food availability and accessibility. However, this assumption continues to be tested with negative results in SA [78]. Many people remain malnourished and poor with limited patterns of food consumption [2,42,73,74,79]. Currently, agriculture constitutes 32% of economic development in SA [71]. Local food production is a growing trend designed to encourage sustainable food consumption patterns among vulnerable communities. Yet, according to findings from recent research articles, many people still go hungry due to limited access to income and food [70,76,78]. This is not surprising because marginalised societies have poor access to land and the resources required for effective food production not only for family consumption purposes but also for business development purposes [64]. Research has established that marginalised black communities own only 4% of the land in SA compared to 72%, which is owned by privileged minority communities who only constitute 9% of the population [43]. Based on this statistical analysis, it is clear why underdevelopment among the marginalised communities remains a big disturbing trend that has remained an unresolved issue in SA [41,77]. This research is supported by Drysdale et al., whose research findings focuses specifically on KwaZulu- Natal [70]. The research findings underscore the significant link between food production capacity, poor food consumption patterns, and poor human and economic development levels emerge. Consequently, these underdevelopment trends are responsible for perpetuating poverty, malnutrition, unemployment, crime,

and poor quality of life among the disadvantaged local communities [4,43,44,75,77,80]. In a recent specific study associated with sustainability and food production, Queenan [42] argued that food production and food consumption systems are facing growing challenges associated with the zero hunger SDG. Furthermore, creating sustainable food production and food consumption patterns to reduce carbon emissions is a trend that is beginning to attract attention in SA [50,81]. Although organic food production and consumption patterns are recognised as pertinent towards the management of the environment, a systematic approach has yet to be fully established in SA [37]. A study on SA by Fourie [81] highlights that the need to align national development plans with sustainability will go a long way towards addressing food security and related health issues.

The research findings, in general, imply that the concept of sustainable food production and food consumption has started attracting attention in SA [42,70], but this is very insignificant because there is hardly any connection between environmental management, human development, and economic development. The need to integrate food production and food consumption in a way that limits food waste, effective management of the ecosystems and related resources while reducing the use of products such as plastics with a focus on localised food production and food consumption is established [37]. Similarly, the application of SDGs to implement, assess and evaluate the levels of sustainability is yet to be fully accomplished in SA, particularly in vulnerable communities [12,35,42,70]. This can be understood based on Maslow's hierarchy of needs [82], which explains why focusing on sustainability practices can only be achieved when the basic needs such as availability and accessibility to food, accommodation, family income, and basic nutrition and health are addressed. The basic needs have yet to be accomplished within marginalised societies in Africa, and food is one of the critical basics.

**Proposition 4.** *Food production and food consumption systems are playing a key role in addressing the human, underdevelopment trends such as food security, poverty, malnutrition, gender equality, lack of income, and unemployability in SA. Food production and food consumption interventions are necessary to bridge the development gap between privileged and marginalised communities in SA.*

### 4.4. Sustainable Food Production and Food Consumption Entrepreneurship

An entrepreneur is perceived from a developmental perspective as an innovator, job creator, game-changer, disruptor, leader, and adventurer with a growth and risk mindset [45]. Interestingly, many research development activists are convinced that sustainable entrepreneurship development could catapult economic, community, and human development within developing economies such as SA [27,38,62,64,71,83–85]. Endorsing these views, other research efforts in SA support this argument and expand the discussion highlighting the fact that entrepreneurship can boost job creation opportunities, income generation and improve the quality of life within marginalised societies [38,41,45,86,87]. Arguing against inequality between the vulnerable and privileged societies in SA, Steyn's research [62] indicates how SMEs as engines of economic development constitute 65–75% of employment in SA among the privileged communities. In agreement with Steyn, other research articles pursue and justify the same critical stance based on strong perceptions of marginalised societies evidenced by poverty and unemployment, which remain as complex problems [8,9,27,47]. Perceived from an economic perspective, a more statistical study indicates that entrepreneurship constitutes more than 40% of GDP in SA [84], demonstrating its potential in advancing economic development levels among the poor. Contrary to these arguments, development critics in SA argue that more than 70% of entrepreneurship business projects launched in the last decade continue to struggle [8,9,40]. Specialised researchers on business entrepreneurship in SA are convinced that enterprise development remains fragmented, linear, and isolated within vulnerable communities [8,9,88]. Further analysis of their research findings suggests that this fragmented approach remains problematic for two main reasons. Firstly, as part of a large supply chain network, a linear approach has a profound negative effect on new business ventures [89,90]. Isolated business functions in

food production often lead to significant failures and wastage of resources [47]. Secondly, it is argued that enterprise development efforts have not engaged successfully with research and intervention to transform the quality of life, especially within marginalised communities [8,91]. Furthermore, a growing school of thought emerging from other research initiatives now suggests that the current controversial and criminal xenophobic events in SA reflect this limited enterprise community engagement with the indigenous marginalised societies [43,62,85,91].

A prescriptive approach to community and economic development tends to focus on big projects and cooperatives. While this may have worked in developed countries such as China and America, it is not working well in SA [78], where a significantly large portion of business ventures fail within the first 2 to 5 years of inception, demonstrating limited viability and sustainability [8,39,62]. A specialised study concurs with this argument concluding that over 70% of SMEs fail within the first 5 years of operation [8,90]. Based on these negative trends, the need for customised and integrated entrepreneurship development in SA is recommended [43,62,64,91]. Factors associated with business development failure in SA include poor managerial and leadership skills implying limited training and development. Research has established that poorly conceptualised and contextualised foreign-driven business intervention trends characterising SA may not succeed, suggesting that knowledge and skills dissemination in business management is a critical success factor [8,39,40,43,45,59,60,62,92]. These studies propose integrated business development and related implementation. A gender-focused study [40] acknowledges an increase in entrepreneurship activity among women in SA. This positive development comes at the back of increased access to financial support now targeting gender-specific entrepreneurship interventions. In contrast, another study [43] argues that a major trend shows 20% of these gendered projects still fail annually despite support and consultancy, training, and development, implying that individualised capacity building needs do not focus only on skills development but also on creating the right business mindset [8,45,92,93]. Consensus regarding the failing business entrepreneurship development trends prevail. A more innovative research-driven approach is needed to address the fragmented, linear, prescriptive, and foreign-based approaches responsible for tremendous failure rates [86]. The link between capacity building, successful entrepreneurship development as a way of bridging the development gap between marginalised and privileged societies is clearly established.

The propositions emerging from this discussion are as follows:

**Proposition 5.** *Although entrepreneurship can address the development gap between marginalised and vulnerable societies in SA, high entrepreneurship failure rates caused by fragmented, prescriptive, and liner approaches, which are foreign-driven, remain as problematic trends limiting the impact on development.*

**Proposition 6.** *Suggestions towards addressing the entrepreneurship failures trends in SA include customised research-driven approach to better understand the context with high levels of capacity building.*

**Proposition 7.** *Entrepreneurship managerial skills are limited in SA with recommendations for capacity building and empowerment to enhance entrepreneurship and bridge the development gap between privileged and marginalised communities.*

### 4.5. Innovative Sustainable and Digital Entrepreneurship

Innovative and transformative digital technologies are becoming the core success factor of integrated and entrepreneurial food production and food consumption [94–98]. Entrepreneurship research initiatives in SA highlight the significance of digital technologies to encourage more creative and innovative food production and food consumption business development [83,86,98,99]. The findings emerging from the research more than justify the use of digital technologies particularly, disruptive technologies, to challenge obsolete and

ineffective traditional ways of business food production, for instance. A wide range of disruptive and transformative digital theories now exist [95,96]. Gonzalez et al. [99] in their research have used some of these theories to identify a wide range of technologies and explore how they can be applied to transform SMEs and lifestyles is SA. These theories confirm that embracing the digital revolution engrained in the proliferation of industry-specific applications as a complex business challenge, is no longer an option but a necessity especially in SA [86,98–100].

Anwana's research [101] suggests the application of innovative digital theories to enhance entrepreneurship in SA. While she reiterates the significance of entrepreneurship in bridging the gap between poverty-stricken rural communities and privileged urban societies, her emphasis is on the development of national policy to reform the area of entrepreneurship. A reflection on the article suggests that a more vigorous practical and integrated entrepreneurs' approach is needed to show how to dump traditional development mindsets, strategies, and business practices and pursue a transformative and innovative business design and implementation. However, the need to contextualise digital entrepreneurship transformation in SA cannot be overemphasised given the current trends showing technological challenges such as availability of smart gadgets and applications, Wi-Fi, and connectivity issues, let alone individual digital capacity, especially in marginalised societies [102–104]. According to Walwyn and Cloete [105], SA is struggling to harness the opportunities for digital development due to political, educational, poor digital managerial skills, and leadership. Consequently, strategic and operational management of digital technologies such as social media, big data, virtual reality, industry-specific applications, robotics, drones, and other more radical forms of artificial intelligence though fundamental in transforming business entrepreneurship, have yet to be effectively harnessed. In contrast, in other countries such as China where digital social media platforms are positively impacting developmental reaction to the COVID-19 pandemic crisis [106]. This is also supported by other researchers who believe capacity building remains a critical digital development factor [86,101,102,107]. SA seems to be doing better in advancing technological development compared to its African counterparts [79,85–109], and yet because of limited contextualisation of technologies, the impact is limited, especially among the poor, vulnerable communities. This is a disturbing trend [39]. The wholesale importation of technologies with limited capacity for effective adaptation to meet the needs of local businesses is a major trend creating a big problem for SA in privileged communities, let alone in marginalised and ghettoised communities [85,100]. While theoretical studies on digital entrepreneurship are documented [83,101,106], specific micro-entrepreneurship theories in marginalised communities in SA have yet to be fully developed. Accordingly, the emphasis should be on innovative adaptations of existing applications and, even more importantly, the development of specific digital applications that can yield contextualised results [39,103]. The goal in SA should focus on digital accessibility and transformation to facilitate innovative and creative businesses development through vibrant start-ups and invigoration of existing enterprises [39,110]. This has even more relevance in the immediate and post COVID-19 pandemic [38,81,82,101,102,106,110,111].

In general, research findings argue that efforts to use innovative and digital entrepreneurship to address the development inequalities existing in SA demand a contextualised approach. Creative ways of harnessing the existing and future digital revolutions will transform lives in SA.

The proposition emerging from the above discussion is:

**Proposition 8.** *Creative and innovative entrepreneurship driven by digital technological accessibility and development will encourage enterprise development in SA among vulnerable communities. This can facilitate bridging the gap between privileged and vulnerable communities in SA.*

## 5. Synthesis of the Literature Reviewed Findings

Research regarding development trends in SA emphasises perpetual inequality between privileged and marginalised communities. Underdevelopment among the vulnerable people in marginalised communities is deep and continues to grow. High levels of poverty, hunger, unemployment, poor economic development, poor levels of income, lack of education, malnutrition, poor health and wellness, gender inequality, criminal activities, poor quality of life characterise the marginalised communities in SA [12,33,34,38]. These issues are linked to sustainability with an emphasis on SDGs. Food production and food consumption entrepreneurship could be used to address the identified SDGs by providing food, employment opportunities, income, economic development, which can then address all the other development issues such as poverty alleviation, health, and upward social mobility [42,49,69,70]. However, entrepreneurship development trends in SA are focusing on privileged societies at the expense of the vulnerable communities [43,62,64,71,83,84]. Currently, a wide range of problems, including fragmentation, prescriptive, and linear and foreign-oriented approaches to enterprise, have seen high failure rates in enterprise development with very limited significance within marginalised societies [1,8,46]. Innovative and creative entrepreneurship based on disruptive technological developmental theories and practices to generate more viable business strategies and related operations could see a drastic change in business growth [46,66,103]. SA will therefore need to deploy an integrated and contextualised approach to yield sustainable entrepreneurship. Meeting the SDGs identified will demand capacity building towards managerial and technological skills. This will empower the vulnerable communities towards the successful enterprise development needed to bridge the development gap between privileged and marginalised communities in SA.

The conceptual ideas emerging from the arguments established are reflected as propositions. These highlight the need to articulate sustainability principles and recommendations for entrepreneurship success, which is currently under impediment through fragmented, prescriptive, linear and foreign conceptualisations [8,39,40,43,45,59,60,62,86,88,92]. The propositions highlight human, social and economic, development gaps between privileged and marginalised communities with an emphasis on the need for contextualised research, theory, and practical business enterprise interventions. The conceptual model (Figure 4) recognises and illustrates an effort to address the identified propositions. A multidisciplinary and integrated approach [87] as a recipe for successful, innovative, digital, sustainable food production and food consumption entrepreneurship with an impact on environmental management, economic, community, and human development is therefore demonstrated in the conceptual model. The concerns advanced regarding fragmentation and prescription of business development interventions [8,39,43,62,64] need to be addressed through the integration of sustainability ideas and practices using a contextualised research-driven approach with a specific focus on marginalised societies. This is the innovative, transformative, and unique part of this model. The model, therefore, emphasises engagement with all the stakeholders at the grassroots level in a customised manner as recommended by scholars [88], including political activists [87] and proponents of development [9,43,64,86]. The model integrates food production and consumption, innovative digital entrepreneurship [39,100], and businesses development and management to address the established SDGs associated with the vulnerable and marginalised communities. Designed to bridge the inequality and development gaps between privileged and marginalised societies, this model is pertinent, especially for developing countries such as SA.

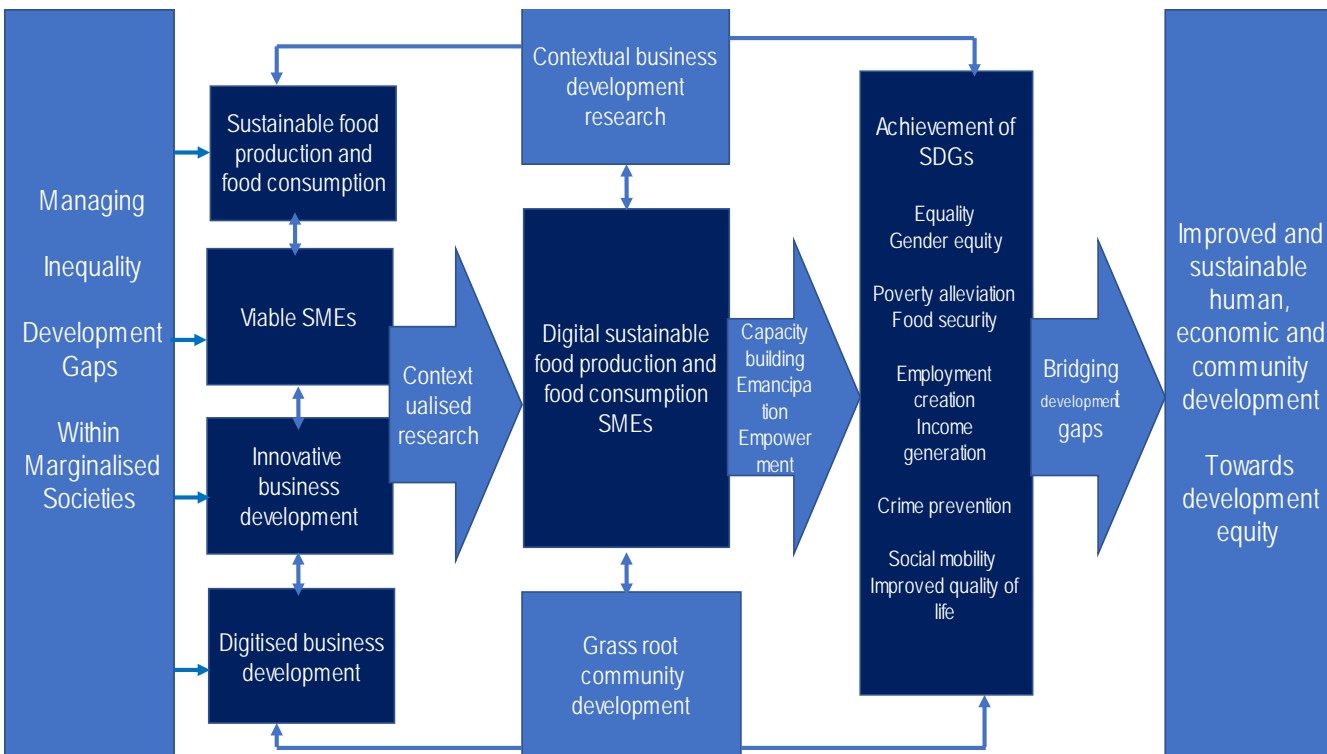

**Figure 4.** Innovative, digitised, and sustainable food production and food consumption entrepreneurship—bridging the development gap between marginalised and privileged societies in SA: a conceptual model.

Embedded in the SDGs are underdevelopment issues such as poverty alleviation, employment creation, criminal activities, income generation, improved quality of life, gender inequity, food insecurity, upwards social immobility. These goals are associated with outcomes of bridging the development gap between privileged and vulnerable and marginalised communities [2,4]. The model, therefore, rejects fragmentation and prescription identified as major causative factors of underdevelopment and development failure [8,47,62,89]. Capacity building towards empowerment and emancipation of the vulnerable members of the community is established as a key development success factor [8,23,62,110]. Training and development will be applied to develop the required knowledge, skills, values, and attitudes for success. Potentially, the challenge implied in this model is the complexity associated with the multidisciplinary, integrated, and customised intervention design and management approach. This demands a diversity of technical, business, research, and leadership skills set at the highest level, which can prove to be challenging in view of stakeholders' interests. The engagement of co-actors is essential to work on collective goals [91]. The unique focus of the model will ultimately facilitate bridging the human, community, and economic development gap between privileged and marginalised communities once the identified SDGs are integrated and contextualised into innovative, transformative, and digitised food production and food consumption entrepreneurship.

## 6. Conclusions

Underdevelopment and inequality trends among the marginalised societies in SA are complex issues consistent with some of the SDGs, including poverty alleviation, employment, generation, income creation, gender equality, food security, no hunger, no crime, upward social mobility, improved quality of life and well-being, human, community, and economic development [8,39,43,62,64]. Although an increase in entrepreneurship development initiatives is recognised, high failure rates associated with fragmented, prescriptive, linear, and foreign-driven approaches towards development are quite disturbing. The

impact of development efforts on vulnerable and marginalised communities remains insignificant. Addressing these issues as a recipe for delivering the identified SDGs, especially the alleviation of poverty and equality [2–4], demanded a research-driven and integrated, contextualised interdisciplinary and systematic approach [93,97], [110,111]. Successful food production and food consumption entrepreneurship can therefore be achieved by effectively integrating with contextualised research, digital technologies, capacity building leading towards innovative and sustainable business development and related practices [72,73,100,105]. Based on these arguments and propositions, a conceptual model (Figure 4) is developed and proposed to facilitate a unique, innovative, and creative sustainable community, human and economic development intervention designed to bridge the development gap between privileged marginalised communities in SA. The model theoretically contributes towards the body of knowledge on sustainable food production and food consumption entrepreneurship within marginalised societies SA. The limitation of the study is that the conceptual framework is yet to be empirically tested. Future research should empirically test this model by validating its applicability and implementation through a systematic pursuit of food production and food consumption entrepreneurship projects within selected marginalised communities in SA. The practical contribution of this framework is a major strength that will open a wide range of research opportunities that could see the establishment of many practical business and community development ventures. Further research could also focus on networking to create opportunities for the viable setting up of a multidisciplinary intervention project consortium designed to engage with the project participants at the grassroots level. Although this customised, collaborative research approach has been suggested across a range of studies focused on viable development project interventions [8,10,87,89], it has yet to be fully explored. The need to generate a strong and grassroots-level analysis of the specific development needs and interests of the vulnerable members of the marginalised societies in question and then engaging with them in the design and development of practical intervention projects based on the framework will shape future research directions. Furthermore, research efforts can create viable criteria for measuring, assessing, and evaluating the success of such development interventions.

**Author Contributions:** Conceptualization, H.R., F.S., J.C., H.W. Methodology F.S. software, F.S. validation, F.S., H.R. and formal analysis, F.S., H.R., J.C., H.W. investigation, F.S., J.C., H.W.; resources, F.S., H.R., H.W., J.C., data curation, H.R., F.S., J.C., H.W.; writing—original draft preparation F.S., J.C., H.W.; writing—review and editing H.R., F.S., G.C.; visualization, F.S.; supervision, H.R., F.S.; project administration F.S. All authors have read and agreed to the published version of the manuscript.

**Funding:** This research received no external funding.

**Institutional Review Board Statement:** Not applicable.

**Informed Consent Statement:** Not applicable.

**Data Availability Statement:** Not applicable.

**Conflicts of Interest:** The authors declare no conflict of interest.

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
