# Peer review of "Innovative and Sustainable Food Production and Food Consumption Entrepreneurship: A Conceptual Recipe for Delivering Development Success in South Africa"

_sustainability, doi:10.3390/su131911049_

Round 1

Reviewer 1 Report

The authors have addressed my review comments. I have outlined a few minor suggestions below. 

Lines 114 - 126 Describe in greater detail what methods you will use to achieve the objectives mentioned (expand on "using extant literature"). 

Reformat the top box in Figure 1 so that it is fully readable. 

Reformat Figure 2 (Some text cut off at top and bottom of boxes). 

Reviewer 2 Report

Thank you for the opportunity to become acquainted with the revised manuscript.
It should be noted that the main drawback of the previous material was the lack of any new findings, other than systematization of known facts on social and economic development in South Africa. The author should have defined the development trends in target-oriented entrepreneurship to enhance the probability of sustainable development of a particular society.
I will dwell on some shortcomings of this version of the manuscript. Section 3.1. Grounded Theory contains material on how to generate new ideas and how to conduct academic research. But it fails to provide any information on the topic of the manuscript. To my mind, having offered the related material, one of the reviewers had meant you to analyze rather than to present the methodology of scientific research in your manuscript. Section 3.2. Methodological Сhoices is rather generalized. As for the presented research methods, they are  obvious but, in my opinion, insufficient. In particular, "... qualitative method using content analysis based on extant literature". The authors note: "The rationale behind this position emanates from the fact that the content analysis was linked to the South African situation." But it goes without saying that the research should be based on information about South Africa. Regarding the final conceptual model, it is simply an integration of sustainable development goals rather than analytical research result, contrary to the assertion of the authors: “The developed conceptual model therefore contextualises and generates an integrated theoretical perception of human, social and economic development issues, sustainable development goals and related food production and food consumption entrepreneurship using a grounded theory approach”.
Generally speaking, the presented Propositions should entail both empirical findings and starting points for reasoning. Nevertheless, the revised manuscript contains detailed and unique information on both regional specifics and the peculiarities of entrepreneurship in the context of food production. In our opinion, it is necessary to extend this trend to any statements of the authors in order the material should acquire the desired uniqueness. 

Reviewer 3 Report

This is an interesting paper and I enjoyed reading it. However, there are essential weaknesses that need to be addressed.

0) Abstract: Authors should state their contribution in terms of issue problems solved or ameliorated, theory or policy dilemmas resolved, or the like. Abstract should offer at least one example of a theoretical or managerial implication that authors concluded after their work.

1) The introductory/opening section should communicate a little clearer the literature gaps, as well as the study's aims & objectives in order to facilitate the flow of the study.

2) It is important to read and cite (where appropriate) current literature, providing a substantial number of citations to support your work. It is also important to read (and, if relevant, cite) papers that have already been published in Sustainability Journal, too. This will help to show the consistency of your research with the debate taking place in the journal.

Additional references to recent & relevant empirical studies could increase the quality of the research paper and provide a much clearer message to the reader - these may help you building your discussion which needs to be extended. Add the following (two references related COVID) to your reference list:

Al-Omoush, K. S., Simón-Moya, V., & Sendra-García, J. (2020). The impact of social capital and collaborative knowledge creation on e-business proactiveness and organizational agility in responding to the COVID-19 crisis. Journal of Innovation & Knowledge, 5(4), 279–288. https://10.1016/j.jik.2020.10.002

Xie, X., Zang, Z., & Ponzoa, J. M. (2020). The information impact of network media, the psychological reaction to the COVID-19 pandemic, and online knowledge acquisition: Evidence from Chinese college students. Journal of Innovation & Knowledge, 5(4), 297–305. https://10.1016/j.jik.2020.10.005

Some of the statements you make are entirely obvious and should be supported in the text by these specific references.  

3) At the end of the ´Conclusion´ section, the author should include clear statements as to where research should now go – what are the issues requiring further research and investigation? The author has to suggest challenges and possible new directions for future work. Perhaps: TO DO AN EMPIRICAL PAPER, TOO.

4) Carefully check the references, so as to make sure they are all complete and follow the Guidelines to Authors.

5) Finally, when you submit the corrected version, please do check thoroughly, in order to avoid grammar, syntax or structure/presentation flaws. Make sure you retain a formal/academic-specific style of presenting your work throughout the text - (if necessary) please seek for professional English proofreading services or ask a native English-speaking colleague of yours in order to refine and improve the English in your paper.

Thank you for the opportunity to read the paper.

Round 2

Reviewer 2 Report

Thank you very much for your manuscript. Now it's good.

Reviewer 3 Report

Authors did not include these two references:

Al-Omoush, K. S., Simón-Moya, V., & Sendra-García, J. (2020). The impact of social capital and collaborative knowledge creation on e-business proactiveness and organizational agility in responding to the COVID-19 crisis. Journal of Innovation & Knowledge, 5(4), 279–288. https://10.1016/j.jik.2020.10.002

Xie, X., Zang, Z., & Ponzoa, J. M. (2020). The information impact of network media, the psychological reaction to the COVID-19 pandemic, and online knowledge acquisition: Evidence from Chinese college students. Journal of Innovation & Knowledge, 5(4), 297–305. https://10.1016/j.jik.2020.10.005

Author Response

Please see attached cover letter.

Round 3

Reviewer 3 Report

Nothing

This manuscript is a resubmission of an earlier submission. The following is a list of the peer review reports and author responses from that submission.

Round 1

Reviewer 1 Report

Title: Innovative Food Production and Food Consumption Entrepreneurship: A Recipe for Delivering Global Sustainable Goals in South Africa

Thank you for the opportunity to review this manuscript. The authors present a review and conceptual framework for achieving SDGs through the food system in SA. This manuscript topic is of interest, but at present the arguments are not clearly linked and could benefit from addressing the suggestions below. 

Additionally, please include line numbers in subsequent revisions to aid reviews in referencing specific text in the manuscript. 

Introduction

Explain what the gender development index is and what the score means in the context of SA and the SDGs. 

It would also be helpful to include an in-depth discussion of the current context and future outlook/importance of food sustainability in SA.

Figure 1

Please order SDGs by ascending number within each of the three pillars or explain why they are not in ascending order?

Check consistency (e.g., #8 Economic is capitalized in Decent work and Economic growth, no space between text and #7)

  1. Literature Review

Overall, the authors’ propositions are not always clearly linked to the context of SA and specifically to food production and consumption. 

2.1  

The authors also mention the three pillars of sustainability, but there is no contextualization of environmental sustainability in this discussion. 

2.2.

Using the term “supreme race” seems to perpetuate use of a racist concept and does not appear warranted in this statement. Additionally, the following sentence needs further explanation and a citation, or should be removed:

The dependence syndrome emanating from the apartheid legacy is significant within many people and organisations leading to a sense of entitlement especially among most black people.

How do you know that there is a “sense of entitlement” among most black people? What is this entitlement in relation to and what is the dependence syndrome referenced? This section needs to be revised and the authors should reflect on how their writing places blame/responsibility on certain communities for their current status.

What is the time period referenced here?

Current employment and income generation trends have seen a downward trend with 3 million people losing their jobs [36].

2.3

It is helpful to define SADC and other acronyms again at first use in the text, even if included in the abstract. 

What is meant here? Wording is unclear. 

Advanced this arguments ascertain that the high levels of food insecurity is challenging to SA demanding immediate action to align with the SDGs.

What is meant by “human community”?

2.4

Who is “his” or is this incorrect? 

In his study [45], demonstrates how SMEs as engines of economic development associated with poverty alleviation, constitute 65%-75% of employment in SA among the privileged.

Do business ventures always fail or is it a large proportion of the time? 

...where business ventures fail within the first 2 to 5 years of inception demonstrating limited viability and sustainability [45].

Conclusions

It is unclear how this conceptual model could and should be empirically tested and validated. 

Reviewer 2 Report

Thank you for the opportunity to get acquainted with this work. The manuscript deals with the socio-economic situation in South Africa, and it has confirmed my idea of ​​this region.
However, the authors did not present any new information. The material systematizes the known facts about this region, which were previously presented in various sources. The authors combine the totality of facts into a concept that, in our opinion, is not new. Its significance lies only in the systematization of known information. Moreover, the provided statements and recommendations could be used by most countries all over the world. I believe that the authors should focus on the specifics of the selected region, and find truly unique features in order to develop peculiar recommendations. 

Reviewer 3 Report

Most importantly you need to strengthen the framing. What gap or unresolved issue in the literature are you seeking to address, and how then will your study provide a unique, value-added contribution to this literature? Your introduction should communicate explicitly how your study will test or advance theory. The lack of theoretical framing in the introduction carries over to your background  -- that is, none of your literature is grounded. To help you with this, I advice you to read these references which provides clear direction on how to set up well-grounded theory:

  • Campbell, J.P. and Daft, C.L. Hulin (1982). What to study: generating and developing research questions. Beverly Hills, CA. Sage.
  • Cummings, L.L., and Frost , P.J. (1995). Publishing in the Organizational Sciences. Thousand Oaks, CA. Sage.
  • Sutton, R.I. and Staw, B.M. (1995). What theory is not. Administrative Sciences Quarterly, 40 (no. 3), pp. 371-384.
  • Feldman, D. (2005). Writing and reviewing as sadomasochistic rituals. Journal of Management, 31 (no. 2), pp. 325-329.

Also, I could not follow major parts of your text because you did not provide sufficient descriptive context. The paper needs to be made more generally accessible to people outside the particular research niche your study focuses on. 

In sum, unless these other fundamental issues are addressed the chances of you publishing this research in an SSCI/SCI listed journal are low.